# Maize-YOLO: A New High-Precision and Real-Time Method for Maize Pest Detection

**DOI:** 10.3390/insects14030278

**Published:** 2023-03-10

**Authors:** Shuai Yang, Ziyao Xing, Hengbin Wang, Xinrui Dong, Xiang Gao, Zhe Liu, Xiaodong Zhang, Shaoming Li, Yuanyuan Zhao

**Affiliations:** 1College of Land Science and Technology, China Agricultural University, Beijing 100083, China; 2Key Laboratory of Remote Sensing for Agri-Hazards, Ministry of Agriculture and Rural Affairs, Beijing 100083, China

**Keywords:** YOLO, maize pest, real-time, object detection, artificial intelligence

## Abstract

**Simple Summary:**

Maize is one of the world’s most important crops, and pests can seriously damage its yield and quality. Detection of maize pests is vital to ensuring the excellent productivity of maize. Traditional methods of pest detection are generally complex and inefficient. In recent years, there have been many cases of plant pest detection through deep learning. In this paper, we propose a new real-time pest detection method based on deep convolutional neural networks (CNN), which not only offers higher accuracy but also faster efficiency and less computational effort. Experimental results on a maize pest dataset show that the proposed method outperforms other methods and that it can balance well between accuracy, efficiency, and computational effort.

**Abstract:**

The frequent occurrence of crop pests and diseases is one of the important factors leading to the reduction of crop quality and yield. Since pests are characterized by high similarity and fast movement, this poses a challenge for artificial intelligence techniques to identify pests in a timely and accurate manner. Therefore, we propose a new high-precision and real-time method for maize pest detection, Maize-YOLO. The network is based on YOLOv7 with the insertion of the CSPResNeXt-50 module and VoVGSCSP module. It can improve network detection accuracy and detection speed while reducing the computational effort of the model. We evaluated the performance of Maize-YOLO in a typical large-scale pest dataset IP102. We trained and tested against those pest species that are more damaging to maize, including 4533 images and 13 classes. The experimental results show that our method outperforms the current state-of-the-art YOLO family of object detection algorithms and achieves suitable performance at 76.3% mAP and 77.3% recall. The method can provide accurate and real-time pest detection and identification for maize crops, enabling highly accurate end-to-end pest detection.

## 1. Introduction

Maize is one of the most important foods and industrial crops in China, but it always faces the threat of pests and diseases during its growth, which leads to a decrease in yield and quality [1]. To avoid this, strict visual monitoring is needed for the early detection of pest and disease infestation of the crop. The traditional way of detection is to identify plant pests and diseases by agricultural experts on-site or by farmers based on their experience [2]. This method is not only time-consuming and labor-intensive but also results in misjudgment due to subjective factors of the inspectors and, thus, the blind use of drugs. This situation will not only bring pollution to the environment but also cause unnecessary economic losses. The application of computer vision methods in the field of plant pest detection has become a research hotspot. The ability to accurately identify the type of pests and locate them is an important aspect of crop production monitoring, as well as the basis for making plant protection prescriptions and automatic and precise application of medicine through images captured by various intelligent vision devices [3].

In recent years, some researchers have used image processing and machine learning techniques to detect crop diseases [4]. However, traditional machine vision methods are less robust in complex scenes, so it is difficult to meet the needs of complex scenes. With the excellent performance of deep learning at the ImageNet Large-Scale Visual Recognition Challenge (ILSVRC), deep learning has been rapidly developed. Among them, deep learning models represented by convolutional neural networks (CNN) have been successful in many fields, such as computer vision [5] and natural language processing [6], and many deep learning architectures are gradually applied to agricultural crops pest identification, such as AlexNet [7], GoogleNet [8], VGGNet [9], ResNet [10], and Vision Transformer [11], which has been introduced from the field of natural language processing to the field of CV in recent years. Object detection of pests is one of the main tasks of plant pest detection, and the aim is to obtain accurate location and class information of pests, which can be well solved by CNN-based deep learning feature extractors and integrated models. Sabanci et al. proposed a convolutional recurrent hybrid network combining AlexNet and BiLSTM for the detection of pest-damaged wheat [12]. Gambhir et al. developed a CNN-based interactive network robot to diagnose pests and diseases on crops [13]. Sun et al. proposed a multi-scale feature fusion instance detection method based on SSD, which improved on SSD to detect maize leaf blight in complex backgrounds [14]. Although all the above studies have suitable accuracy, they cannot meet the demand for real-time object detection.

In computer vision, real-time object detection is a very important task. YOLO is a popular family of real-time object detection algorithms. The original YOLO object detector was first released in 2016 [15]. This architecture is much faster than other object detectors and has become the latest technology for real-time computer vision applications. Currently, YOLO has been widely used in plant pest identification. For example, Roy et al. proposed an improved YOLOv4-based real-time object recognition system, Dense-YOLOv4, by integrating DenseNet into the backbone to optimize the transmission and reuse of features [16]. Using the improved PANet to acquire location information and detect mango growth in complex scenes in orchards. Lawal et al. proposed YOLO-Tomato-A and YOLO-Tomato-B models based on YOLOv3 to detect tomatoes in complex environments and improved the performance of the models by adding labeling methods, dense architecture, spatial pyramid pooling, and the mish activation function to the YOLOv3 model [17]. Li et al. proposed an architecture for plant pest and disease video detection based on a combination of deep learning and custom backbone networks [18]. Experiments demonstrated that the customized DCNN network has outstanding detection sensitivity for rice stripe wilt and rice stem borer symptoms compared with YOLOv3 with VGG16, ResNet50, and ResNet101 as the backbone. Zhang et al. combined the pooling of spatial pyramids with YOLOv3 to achieve inverse by combining upsampling and convolution operations. Convolution can effectively detect small-sized plant pest samples in images, and the average recognition accuracy can reach 88.07% [19]. Tian et al. proposed a deep learning-based method for Apple Anthracnose damage detection, which optimized the low-resolution feature layer of the YOLOv3 model by DenseNet and greatly improved the utilization of neural network features and the detection results of the model [20]. In summary, YOLO can achieve end-to-end detection by converting the object detection task into a regression task and using global features to directly classify and localize the object, which greatly improves the speed of detection. However, it is difficult for the YOLO family of object detection algorithms to achieve a suitable balance between accuracy, speed, and computational effort at the same time.

With the YOLO family of object detection algorithms continuously updated, YOLOv7 was created in 2022 [21], which claimed that YOLOv7 is the fastest and most accurate real-time detector to date. There is less research on the application of YOLOv7 to plant pest detection, so we tried to apply YOLOv7 to plant pest detection while we further optimized its accuracy, speed, and computational load. On the other hand, although some researchers have explored DL-based detection and identification of pests, there are fewer studies on pest detection and identification for maize, an important agricultural crop. In this study, we propose the Maize-YOLO algorithm for maize pest detection. The proposed algorithm achieves a suitable balance between accuracy, computational effort, and detection speed to achieve fast and accurate detection of pests. The main contributions of this work are as follows:

(1) We inserted the CSPResNeXt module into the YOLOv7 network, replacing the original ELAN module, to improve accuracy while reducing model parameters and computational effort.

(2) We replaced the ELAN-W module in YOLOv7 with the VoVGSCSP module, which reduces the complexity of the model while maintaining the original accuracy and improving the speed of detection.

(3) We tested the detection of highly damaging pests on maize using Maize-YOLO on the IP102 dataset. We achieved 76.3% mAP and 77.3% Recall with a substantially reduced computational effort and an FPS of 67. better than most of the current classical detection algorithms, including YOLOv3, YOLOv4, YOLOv5, YOLOR, YOLOv7. and we conducted an ablation experiment to separately analyze the effects of CSPResNeXt-50 and VoVGSCSP on Maize-YOLO.

## 2. Materials and Methods

### 2.1. Maize-YOLO

Since the release of YOLO, many versions based on the YOLO architecture have been derived, such as Scaled-YOLOv4 [22], YOLOv5, YOLOR [23], etc. YOLOv7 is the latest version of the YOLO series and is currently the fastest and most accurate real-time object detection model for computer vision tasks. Compared to previous versions, YOLOv7 greatly improves real-time object detection accuracy without increasing inference costs. It can generate models at different scales to meet different inference speed requirements by means of composite scaling methods, such as YOLOv7x, YOLOv7-w6, YOLOv7-e6, and YOLOv7-d6.

The network structure of YOLOv7 is mainly composed of CBS, UPSample, MP, SPPCSPC, ELAN, REPConv, etc. The structure of each module is shown in Figure 1. The CBS module with a kernel of 3 and stride of 1 is mainly used to extract features, and the CBS module with a kernel of 3 and stride of 2 is mainly used to downsample. Upsample and MP are upsampling and downsampling modules, respectively, which are used to consider the maximum and local value information of local regions. SPPCSPC is a modified spatial pyramidal pooling structure in which ELAN and ELAN-W are efficient network structures that regulate the network to be able to learn more features by controlling the shortest and longest gradient paths. REPConv modules are of two types, used for train and deployment, respectively.

The basic framework of Maize-YOLO can be divided into three parts: Input, Backbone, and Head. Input is responsible for operations such as image Mosaic data enhancement, adaptive anchor frame calculation, and adaptive image scaling. Backbone consists mainly of ELAN, CBS, and CSPResNeXt-50 modules. It uses the CBS base convolution module for feature extraction and the CSPResNeXt-50 module to reduce MAC and achieve the most efficient computation. Head aggregates the image features by using the SPPCSPC and ELAN modules, using the VoV-GSCSP module to minimize spatial complexity while maximizing the loss of some of the semantic information caused by spatial compression and channel expansion of the feature map. Finally, the Rep module adjusts the channels of the output features, which are then combined with 1 × 1 convolutional layers for prediction and output. Overall, Maize-YOLO provides a faster and more powerful network architecture, provides more effective feature integration methods, more accurate object detection performance, better loss functions, and higher label allocation and model training efficiency. Therefore, Maize-YOLO can be trained faster on small datasets without any pre-training weights. We have made the following major changes: (1) replace the ELAN-W module with the VoVGSCSP module in the YOLOv7 framework; (2) replace some ELAN modules with CSPResNeXt-50 modules. The network structure is shown in Figure 2.

### 2.2. CSPResNeXt-50

CSPNet (CrossStagePartialNetworks) is a lightweight network structure proposed in 2020 that combines feature maps of the underlying layers into two parts and a proposed cross-stage hierarchy to achieve richer gradient combinations while reducing model computations [24]. CSPNet, on the other hand, can be easily applied to architectures such as ResNeXt, ResNet, and DenseNet. Therefore, this study combines CSPNet and ResNeXt to form CSPResNeXt to replace some ELAN modules in YOLOv7. The purpose of this study is to improve the accuracy of our model while alleviating the problem that much inference calculation needs to be performed from the perspective of network architecture. The CSPResNeXt-50 framework is shown in Figure 3. Since only half of the feature channels pass through ResUnit, there is no need to introduce the bottleneck layer anymore. When FLOPs are fixed, the memory access cost (MAC) can be lowered to its lower limit.

### 2.3. VoVGSCSP

GSConv is a new lightweight convolution method proposed in 2022 [25]. The structure, shown in Figure 4, enhances the non-linear representation by adding DSC layers and a shuffle, which preserves the hidden connections between each channel as much as possible with less time complexity. This method makes the output of the convolution calculation as close as possible to that of the SC (channel-dense convolution) and reduces the calculation cost. If GSConv is used in the backbone, the network layer of the model will be deeper, which will increase the resistance of the data flow and greatly increase the reasoning time. When GSConv is used in the neck, since the channel size of the feature map is the largest and the width and height dimensions are the smallest, using GSConv to process the feature map in series can reduce duplicate information and does not require compression.

VoVGSCSP is a continuous introduction of GS bottleneck based on GSConv, and then a cross-stage partial network (GSCSP) module, VoVGSCSP, is designed by using the one-shot aggregation method, as shown in Figure 5. The module balances the accuracy and speed of the model well, reducing computational and network structure complexity while maintaining sufficient accuracy and a high reuse rate of extracted features.

### 2.4. Dataset

IP102 is a large-scale pest identification dataset covering 102 common crop pests [26]. As IP102 contains images of the entire pest life cycle, the larval and adult stages of some pests are often very different in appearance, and some different types of pests are similar to each other. In addition, the background of the images is complex, and some pests are similar in color to the background. The combination of these factors makes the identification of the IP102 dataset challenging. In this study, only pest species that are damaged by maize were targeted, as shown in Table 1, with 13 species. They are referred to as IP13 in this paper.

### 2.5. Performance Evaluation Methods

In this study, mean average precision (mAP), floating-point operations (FLOPs), and frames per second (FPS) are selected as evaluation indicators. mAP represents the average value of AP, which is used to measure the overall detection accuracy of object detection. For object detection, AP and mAP are the best indicators to measure the detection accuracy of the model [27]. FLOPs represent the amount of computation and are used to measure the complexity of the model. FPS indicates how many images can be detected by the network per second as an indicator of object detection speed. The formula is shown in Equations (1)–(4):(1)$$\mathrm{Precision}\left(P\right)=\frac{\mathrm{TP}}{\mathrm{TP}+\mathrm{FP}}$$
(2)$$\mathrm{Recall}\left(R\right)=\frac{\mathrm{TP}}{\mathrm{TP}+\mathrm{FN}}$$
(3)$${\mathrm{AP}}_{i}=\int_{0}^{1}P_{i}\left(R_{i}\right){\mathrm{dR}}_{i}$$
(4)$$\mathrm{mAP}=\frac{1}{C}\overset{C}{\underset{i=1}{\sum }}{\mathrm{AP}}_{i}$$

Among them, Precision reflects the correctness of the model in all detected boxes. Recall represents the coverage of the detection frame to all ground truths predicted by the model. AP evaluates the performance of the model in each category by considering P and R indicators. The value of AP is equal to the area between the P–R curve and the coordinate axis. TP (true positive) indicates the number of correctly predicted positive instances, and FN (false negative) indicates the number of incorrectly classified positive instances; FP (false positive) is the number of negative instances classified as positive categories. C is the number of pest species and represents the precision of the ith pest category. $R_{i}$ represents the recall rate of the ith pest category.

## 3. Experimental Results and Discussion

Before model training, this paper divides the dataset into a training set and a validation set in a ratio of about 8:2 for 5-fold cross-validation, where the training set is 3600 images, and the test set is 932 images. This means that the whole dataset is divided into 5 equal parts, each of which accounts for about 20% of the whole dataset. We trained the model on 4 parts and validated it on the remaining 1 part. This operation was repeated 5 times and took the average value as the final result. To ensure the originality of the dataset, we only scaled the input images adaptively, and since images with too large a size or resolution can significantly increase the detection time of the model, we scaled the input images to 3,640,640. All experiments were conducted on a server configured with 16 Intel (R) Xeon (R) Gold 5218 CPU and 2 NVIDIA GeForce RTX 3090. These GPUs were deployed based on Python 3.9 under Linux Ubuntu 20.04 LTS operating system. Our model is built under the Pytorch 1.11.0 deep learning framework, and the Adam algorithm is used to optimize the model in the training process. The default super parameter settings are as follows: epoch is 300, batch size is 32, and the initial learning rate is 0.01. The momentum factor and weight falloff are set to 0.937 and 0.0005, respectively. In this study, we initialize the parameters of all YOLO model backbone networks using pre-trained weights.

### 3.1. Detection Results of Maize-YOLO

Based on the IP13 dataset, we compared Maize-YOLO with other models that are currently performing well, including RetinaNet, Faster R-CNN, YOLOv3, YOLOv4, YOLOv5, YOLO-Lite, YOLOR, YOLOv7, and different versions of these algorithms. The backbone of RetinaNet and Faster R-CNN is ResNet50. We first analyzed the detection results of Maize-YOLO, as shown in Figure 6. The loss curve of Maize-YOLO stabilizes later in the model training process, with the loss values fluctuating only within a small range.

On the other hand, we obtained the P–R curves for each species. According to Figure 7 and Table 2, we can see that Maize-YOLO successfully identified most species of pests, with mAPs that could reach 76.3% and the highest AP even reaching 99.7%, but the identification of category 17 (White margined moth), category 18 (Black cutworm), category 19 (Large cutworm), and category 20 (Yellow cutworm) were generally identified because the dataset contained both larval and adult stages of these four categories of pests, and their different growth periods differed greatly in appearance, which caused great difficulties in identification.

### 3.2. Comparison of Other Classical Models

As shown in Table 3, the model size of Maize-YOLO is 33.4M, the FLOPs are 38.9G, and the mAP reaches 76.3%. Its mAP exceeds the mAP of almost all other classical detection models (except YOLOR-E6). In addition, some YOLO models with similar mAP size as Maize-YOLO, such as YOLOR-D6, YOLOR-E6, YOLOR-P6, YOLOv7-E6E, which are two to six times more computationally intensive than Maize-YOLO. On the other hand, Maize-YOLO achieves an FPS of 67, and its detection speed surpasses that of most YOLO models (except for YOLO models with less than 20G of computation), which indicates that our method can meet the requirements of real-time detection. In conclusion, our model strikes a suitable balance between speed, accuracy, and computational effort.

### 3.3. Ablation Experiment

To distinguish the respective characteristics of the CSPResNeXt-50 module and the VoVGSCSP module, we performed ablation experiments with the method proposed in this study, and the results are presented in Table 4. We added the two separately to the YOLOv7 network and analyzed the impact of each module on YOLOv7 and Maize-YOLO. 

(1) Effects of CSPResNeXt-50, Compared to YOLOv7, YOLOv7+CSPResNeXt-50 has 4.8% fewer parameters, 61% less computation, 4% more mAP, but a decrease in FPS by 8. This demonstrates that the CSPResNeXt-50 module, when replacing the original ELAN-W module, does help to significantly reduce computational costs. By using a cross-stage splitting and merging strategy, the CSPResNeXt-50 module is able to effectively reduce the possibility of duplication in the information integration process and also has a suitable improvement in the learning capability of the network.

(2) Effects of VoVGSCSP, compared to YOLOv7, VoVGSCSP has 12% fewer parameters, 11% fewer calculations, 4.5% more mAP, and 14 more FPS. This demonstrates that the VoVGSCSP module can improve the inference speed and mAP of the network without increasing the computational effort and that the depth-wise separable convolution in the VoVGSCSP module can achieve results close to those of normal convolution and is more efficient. We took into account the characteristics of the VoVGSCSP module and applied it to the neck, but we did not replace all ELAN-W modules with VoVGSCSP modules because too many VoVGSCSP modules would lead to a deepening of the network layers, which would increase the resistance of the data flow and significantly increase the inference time. Therefore, in the Maize-YOLO network structure, the feature maps processed with VoVGSCSP modules contain less redundant repetitive information and do not require compression.

(3) The effects of both together, We conducted a series of experiments and found that the CSPResNeXt-50 module made a far greater contribution to reducing computational effort than the VoVGSCSP module. The VoVGSCSP module contributes more to improving the speed of model detection than the CSPResNeXt-50 module. Both have similar improvements to the model’s mAP, and finally, Maize-YOLO combines the best of each with a 10% reduction in the number of parameters, a 63% reduction in computation, a 5.5% improvement in mAP, and a 10 improvement in FPS compared to YOLOv7. Figure 8 shows the confusion matrix for YOLOv7, YOLOv7+CSPResNeXt-50, YOLOv7+VoVGSCSP, and Maize-YOLO. As can be seen from Figure 8, Maize-YOLO combines the features of both CSPResNeXt-50 and VoVGSCSP and has a suitable improvement in recognition accuracy for most types of pests compared to YOLOv7.

In conclusion, Maize-YOLO is not only accurate but also fast enough to meet the requirements of real-time detection. Whether it is used to identify the pest category or to track the movement of the pest, Maize-YOLO is very advantageous in different practical applications. Figure 9 shows some of the results of Maize-YOLO’s detection of pests. It can be seen that Maize-YOLO also performs well in detecting smaller pests, demonstrating the reliability of our model.

## 4. Conclusions

In this study, we propose a real-time intelligent maize pest detection method (Maize-YOLO) that can well balance the relationship between accuracy, speed, and computational effort and outperforms current state-of-the-art real-time object detection algorithms. Our Maize-YOLO differs from current related research [28,29,30] in that it achieves a high level of detection accuracy while maintaining high speed. The method provides accurate pest detection and identification not only for maize crops but also for other crops, enabling end-to-end real-time pest detection.

To maintain the originality of the large-scale pest dataset IP102, we only size-scaled the dataset without excessive pre-processing of the dataset, and the detection results from the model did not show that the unbalanced data resulted in the model being more biased toward those categories with more training samples. In addition, the sample distribution of the dataset reflects the true situation in the natural field environment, with some types of pests occurring more frequently and others being less common. Currently, there are few open-source datasets related to plant pest detection, and we can maximize the use of existing datasets by using data augmentation techniques or GAN to perform richer manual processing on the existing datasets.

There are still some limitations in this study, as some types of pests have a large difference in appearance between the larval and adult stages, which can lead to poor recognition of such types of pests by the model. In the next step of our research, we will refine the model’s identification process by classifying different pest types according to their growth period and morphological differences.

In the future, Maize-YOLO can be combined with web interfaces, intelligent patrol robots, and smartphone applications to achieve an easy-to-operate human–computer interface through the use of an AI-based pest detection system.

## Figures and Tables

**Figure 1 insects-14-00278-f001:**
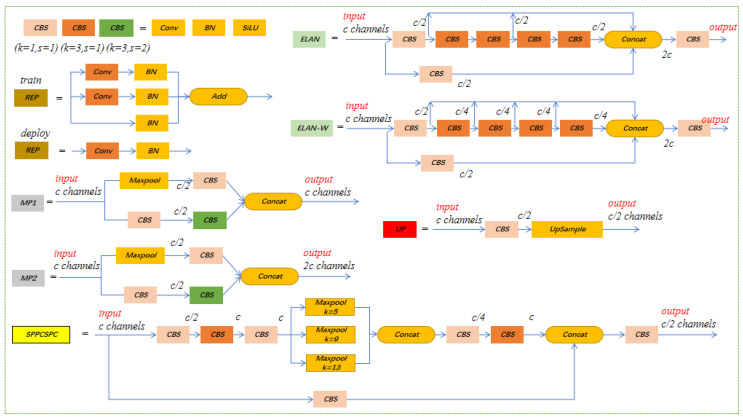
Architecture of several key modules in YOLOv7 network.

**Figure 2 insects-14-00278-f002:**
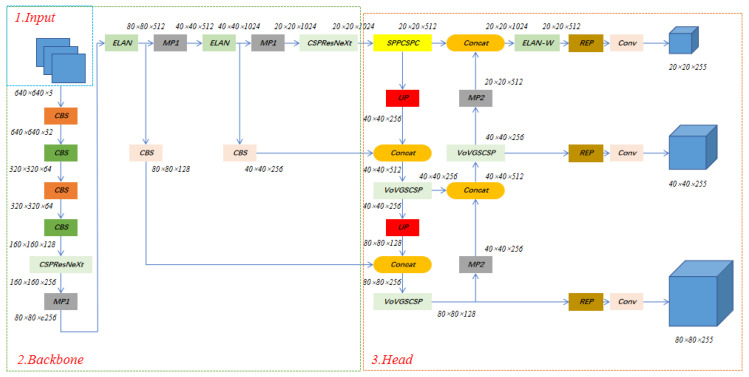
The architecture of Maize-YOLO.

**Figure 3 insects-14-00278-f003:**
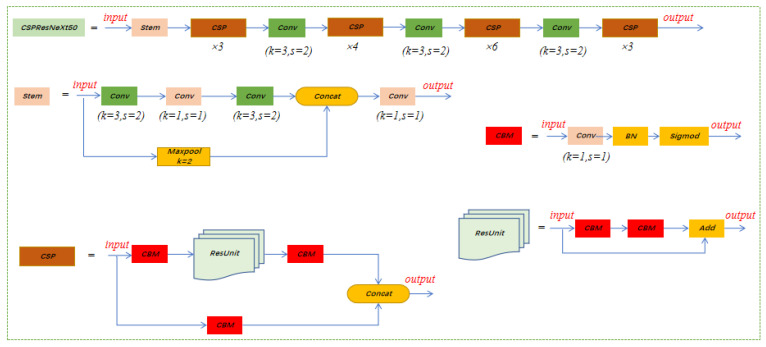
Network structure of CSPResNeXt-50.

**Figure 4 insects-14-00278-f004:**
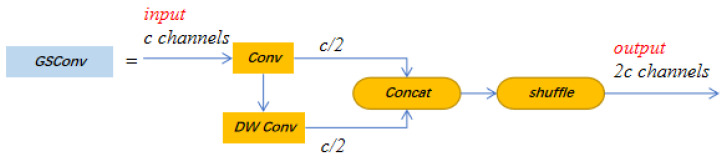
Network architecture of GSConv.

**Figure 5 insects-14-00278-f005:**
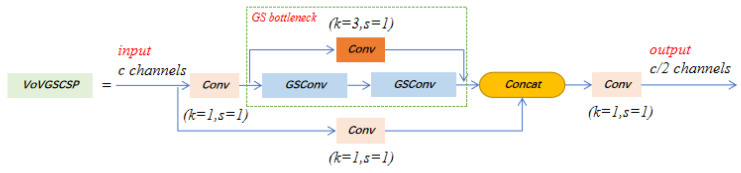
Architecture of VoVGSCSP.

**Figure 6 insects-14-00278-f006:**
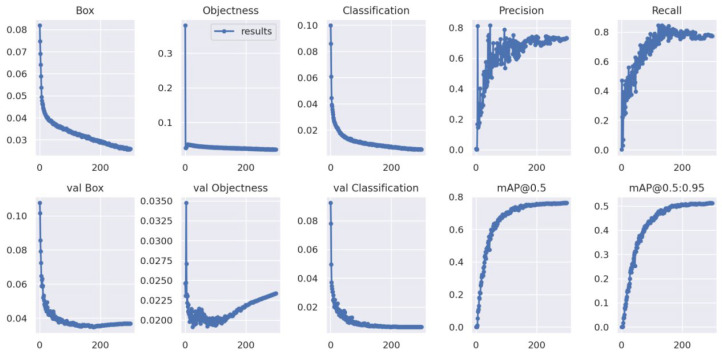
Maize-YOLO curves during training, Box represents the mean of GIoU Loss, Objectness represents the mean of object detection loss, Classification represents the mean of classification loss, val Box represents the mean of GIoU loss in the validation set, val Objectness represents the mean of object detection loss in the validation set, val Classification represents the mean of classification loss in the validation set.

**Figure 7 insects-14-00278-f007:**
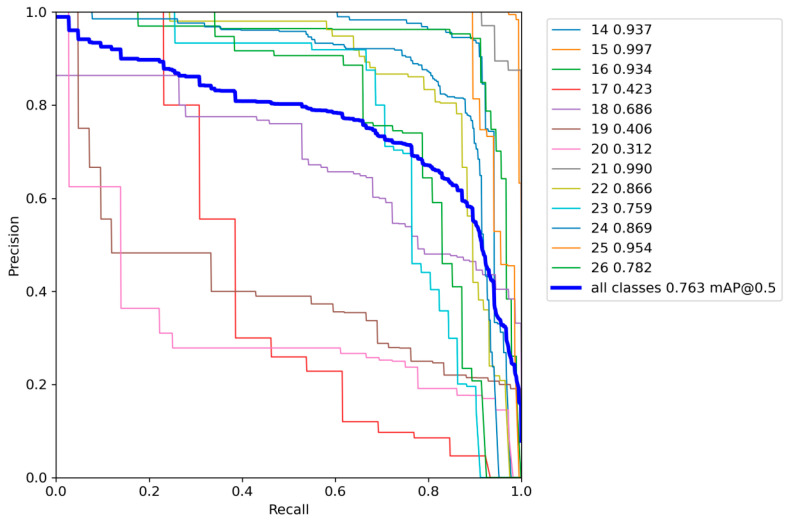
P–R curves and AP values for each type.

**Figure 8 insects-14-00278-f008:**
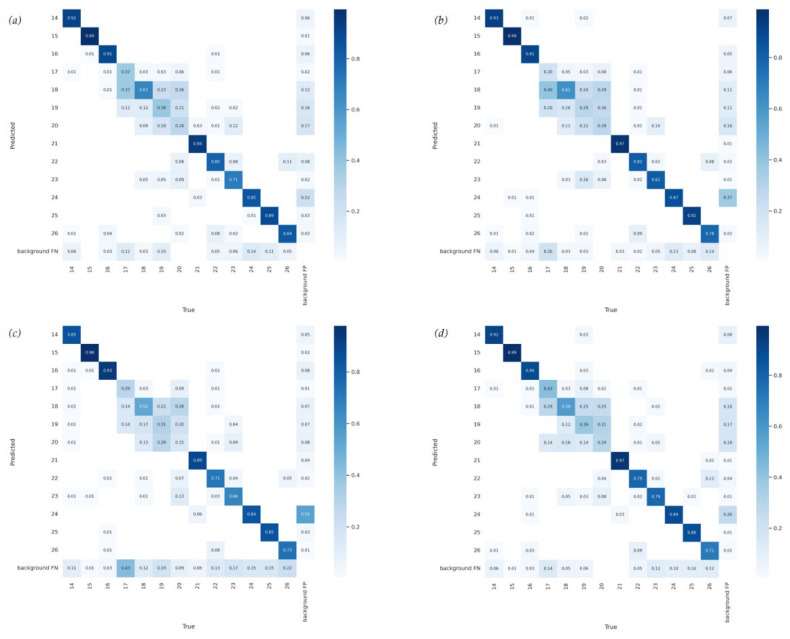
Confusion matrix for Maize-YOLO and its ablation studies. (**a**) YOLOV7+CSPResNeXt-50; (**b**) YOLOV7+VoVGSCSP; (**c**) YOLOv7; (**d**) Maize-YOLO.

**Figure 9 insects-14-00278-f009:**
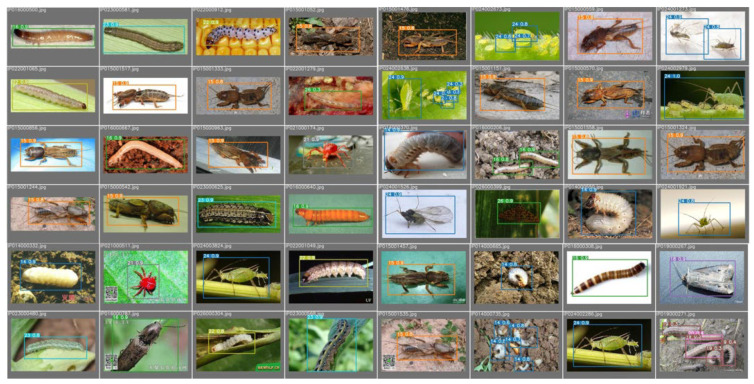
Example of Maize-YOLO results for pest detection.

**Table 1 insects-14-00278-t001:** Pest species and sample numbers in IP13.

Id	Pest Species	Number
14	Grub	436
15	Mole cricket	868
16	Wireworm	424
17	White margined moth	49
18	Black cutworm	300
19	Large cutworm	153
20	Yellow cutworm	194
21	Red spider	160
22	Corn borer	425
23	Army worm	206
24	Aphids	874
25	Potosiabre vitarsis	203
26	Peach borer	241
Total		4533

**Table 2 insects-14-00278-t002:** Detection performance of Maize-YOLO on IP13 datasets for various types of pests.

Class	Images	Labels	P	R	mAP@0.5	mAP@0.5:0.95
14	932	154	88.0%	91.6%	93.7%	51.9%
15	932	181	98.7%	98.9%	99.7%	53.9%
16	932	91	84.5%	92.3%	93.4%	69.6%
17	932	13	54.6%	38.5%	42.3%	35.8%
18	932	72	54.4%	73.6%	68.6%	56.0%
19	932	42	38.5%	54.8%	40.6%	30.9%
20	932	36	27.8%	61.1%	31.2%	24.7%
21	932	35	92.0%	94.3%	99.0%	65.2%
22	932	86	78.1%	86.9%	86.6%	56.2%
23	932	51	86.9%	68.6%	75.9%	48.4%
24	932	257	76.7%	89.1%	86.9%	49.1%
25	932	67	95.2%	89.6%	95.4%	67.0%
26	932	47	77.0%	66.0%	78.2%	56.6%

**Table 3 insects-14-00278-t003:** Performance comparison of object detection algorithms on the IP13 including Maize-YOLO, mAP@0.5 is the average accuracy of all categories when the representative accuracy assessment IoU threshold is 0.5, mAP@0.5: 0.95 is the average accuracy of IoU thresholds weighted from 0.5 to 0.95 in steps of 0.05. Time indicates the training time of the model.

Model	Params	Layer	P	R	mAP@0.5	FLOPs	FPS	mAP@0.5:0.95	F1	Time (h)
RetinaNet	36.6M	152	72.2%	56.53%	65.9%	167.7G	38	39.5%	61.8%	14.549
Faster R-CNN	136.9M	40	48.2%	59.9%	55.0%	402.0G	24	29.0%	46.5%	39.214
YOLOv3	61.6M	333	69.7%	72.2%	71.7%	155.5G	58	45.4%	70.9%	13.796
YOLOv3-SPP	62.6M	342	73.2%	78.3%	74.5%	156.3G	51	48.7%	75.7%	14.196
Scaled-YOLOv4	52.6M	513	58.3%	63.5%	63.3%	119.9G	51	33.9%	60.8%	15.428
YOLOv5-L	46.2M	468	65.5%	55.8%	59.0%	108.4G	45	31.4%	60.3%	12.004
YOLOv5-S	7.1M	270	51.4%	48.2%	49.9%	16.0G	94	23.9%	49.7%	5.663
YOLOv5-N	1.8M	270	47.2%	53.0%	52.6%	4.3G	114	26.8%	49.9%	4.755
YOLOv5-M	20.9M	369	58.6%	69.4%	65.1%	48.4G	52	38.1%	63.5%	8.581
YOLOv5-X	86.3M	567	71.8%	61.2%	65.5%	204.9G	48	38%	66.1%	17.670
YOLO-Lite	4.4M	319	49.3%	55.7%	56.2%	8.8G	178	32.1%	52.3%	5.176
YOLOR-CSP	52.6M	521	65.9%	67.9%	66.9%	119.9G	50	38.4%	66.9%	13.648
YOLOR-D6	151.1M	956	75.8%	77.2%	75.8%	233.3G	29	50.5%	76.5%	26.889
YOLOR-E6	115.3M	740	76.1%	77.0%	76.7%	170.0G	34	51.4%	76.5%	22.324
YOLOR-P6	36.9M	660	70.1%	82.4%	76.2%	80.8G	43	50.9%	75.7%	11.140
YOLOR-W6	79.4M	660	76.3%	72.9%	75.1%	112.6G	41	50.1%	74.6%	16.371
YOLOv7	37.2M	407	69.0%	71.1%	70.8%	105.3G	57	45.1%	70.0%	11.263
YOLOv7-tiny	6.1M	263	66.9%	68.9%	67.7%	13.3G	204	39.5%	67.9%	5.739
YOLOv7-tiny-silu	6.0M	255	69.0%	63.9%	68.1%	13.3G	182	41.1%	66.3%	5.809
YOLOv7-D6	133.0M	702	72.8%	77.6%	74.7%	174.5G	41	50.6%	75.1%	23.574
YOLOv7-E6	110.6M	645	75.6%	77.0%	74.9%	144.9G	44	50.5%	76.3%	22.285
YOLOv7-W6	81.2M	477	75.9%	74.9%	73.3%	102.9G	61	49.7%	75.4%	18.723
YOLOv7-X	70.9M	459	74.4%	67.2%	71.5%	189.1G	42	47.2%	70.6%	16.338
YOLOv7-E6E	151.1M	1032	74.5%	79.0%	76.2%	210.1G	24	51.9%	76.7%	28.158
Maize-YOLO	33.4M	527	73.3%	77.3%	76.3%	38.9G	67	51.2%	75.2%	10.598

**Table 4 insects-14-00278-t004:** Ablation study of Maize-YOLO on IP13.

Model	Param	Layer	P	R	mAP@0.5	FLOPs	FPS	mAP@0.5:0.95	F1	Time (h)
YOLOv7	37.2M	407	69.0%	71.1%	70.8%	105.3G	57	45.1%	70.0%	11.263
YOLOv7+ CSPResNeXt-50	35.4M	434	75.0%	71.7%	74.8%	41.3G	49	50.4%	73.3%	10.689
YOLOv7+ VoVGSCSP	32.7M	515	73.3%	79.1%	75.3%	93.6G	71	49.5%	76.1%	10.942
Maize-YOLO	33.4M	527	73.3%	77.3%	76.3%	38.9G	67	51.2%	75.2%	10.598

## Data Availability

The code and data sets in the present study may be available from the corresponding author upon request.

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
