# Peer review of "Maize-YOLO: A New High-Precision and Real-Time Method for Maize Pest Detection"

_insects, 2023, doi:10.3390/insects14030278_

Round 1

Reviewer 1 Report

In this paper, a high-precision and real-time detection method for corn pest detection is proposed. The CSPResNeXT-50 module and VoVGSCSP module are introduced, which not only reduces the calculation of the model, but also improves the detection accuracy and speed of the network.The article is relatively clear, the experiment is sufficient and the chart is clearly expressed.However, in the method discussion part, there is too much discussion on the basic model yolov7. It is suggested to focus on the innovation part of this paper and the optimization brought by the innovation part

Reviewer 2 Report

1 The authors claim that they reduced the computational resource consumption, how did you measure this?

2 The authors should have emploed cross-valiation for the evaluation method.

3 Are the reported results for validation? The evaluation method does not mentioned testing in data division.

4 How many anchors and what was the location of detection heads?

5 The choice of parameters and evaluation methods should be qualified with appropriate references, see similar studies utilizing Yolo can be cited so that to established the trustworthiness of the models and can provide reliabilit to baseline settings, see Detection of K-complexes in EEG signals using deep transfer learning and YOLOv3. Cluster Comput (2022). https://doi.org/10.1007/s10586-022-03802-0

6 Recent research in plant disease identification (e.g., corn, tomatoes, grapes, apples) especially using transfer learning should be contrasted with your work, why do you need object detection instead of classification?'

7 The reference numbers should be spaced from the text. 

8 The citation numbers in the text need to be improved. Font size is bigger that the rest of the text. 

9 The table of abbreviations is missing but required by the journal template.

Reviewer 3 Report

The authors propsed Maize-YOLO model for the pest species of maize in IP102.  The experimental results show that our method outperforms the current state-of-the-art YOLO family of object detection algorithms and achieves good performance at 76.3% mAP and 77.3% recall. 

However, there are several questions the authors need to explain:

1. The experiments are conducted based on the IP102 dataset. This dataset is designed for pest classification and there is only limited targets on the images. So I think it is not one proper dataset for your detection model. We have multi tiny pests in the actual maize images. The authors have to explain the reason why you choose this dataset or consider other proper dataset.

2. The authors modified YOLOv7 with CSPRResNeXt-50 and VoVGSCSP module. The ablation study is limited in the YOLOv7 model. The comparison study is limited to YOLO models. So did the authors compare other classical detection model on the dataset? 

3. I noticed one ELAN and one ELAN-W module are remained in the Maize-YOLO model in Figure 3 compared with YOLOv7 in Figure 1. Can you explain your reason?

4. The structure and the writing of the paper need to be polished. I don't think it necessary to mention Figure 2 in 2.1. The class ID could just start from 1. For the categories of relatively low identification rate, Is the detection rate also decrease?

Round 2

Reviewer 2 Report

The authors addressed my comments. 

Reviewer 3 Report

The authors answered the questions clearly and the paper was polished based on the reviewer's suggestion.